



# The MESSy DWARF (based on MESSy v2.55.2)

Astrid Kerkweg[1,2], Timo Kirfel[1,2], Duong H. Do[1], Sabine Griessbach[2,3], Patrick Jöckel[4], and Domenico Taraborrelli[1,2]

[1]Institute of Energy and Climate Research 8, Troposphere, Forschungszentrum Jülich, Jülich, Germany
[2]Centre for Advanced Simulation and Analytics (CASA), Forschungszentrum Jülich, Jülich, Germany
[3]Jülich Supercomputing Centre, Forschungszentrum Jülich, Jülich, Germany
[4]Deutsches Zentrum für Luft- und Raumfahrt (DLR), Institut für Physik der Atmosphäre, Oberpfaffenhofen, Germany

**Correspondence:** Astrid Kerkweg (a.kerkweg@fz-juelich.de)

**Abstract.**

Adaptation of Earth system model (ESM) codes to modern computing architectures is challenging, as ESMs consist of a multitude of different components. Historically grown and developed by scientists rather than software engineers, the codes of the individual components are often interwoven, making the optimisation of the ESMs on modern computing architectures

rather challenging, if not impossible.

Thus, in the last years the codes became increasingly modularised and with that, different components are disentangled from each other. This helps porting the code section by section to modern computing architectures, e.g. to GPUs.

Since more than 20 years, the modularisation is the fundamental concept of the Modular Earth Submodel System (MESSy). It is an integrated framework providing data structures and methods to build comprehensive ESMs from individual components.

Each component is coded as an individual, so-called submodel. Components can be individual process implementations, e.g. a cloud microphysical scheme, a convection scheme, dry deposition of tracer gases, or diagnostic tools, e.g. output on a profile station location, on (flight) trajectories, or on satellite orbits. Each submodel is connected via the MESSy infrastructure with all other components, together forming a comprehensive model system. MESSy was mainly developed for research in atmospheric chemistry, and so far it is always connected to a dynamical (climate or weather forecast) model, what we call basemodel. The

basemodel is a development outside the MESSy framework. However, running a full dynamical model for technical tests when porting only one submodel is a tedious task and unnecessarily resource consuming. Especially, as for such technical tests a simple grid, parallelisation scheme, and time control are sufficient in many cases.

Therefore, we developed the so-called MESSy DWARF, a simplified basemodel based on the MESSy infrastructure. We implemented the definition of a very simple grid, parallelisation scheme, and a time control to replace a fully-fledged basemodel.

The MESSy DWARF can not only be used for technical applications, such as porting individual component implementations to GPUs, but it is also applicable for scientific purposes running simplified models (with only a selection of submodels), e.g., a chemical box model for the analysis of chamber experiments. In this paper we introduce the technical setup of the MESSy DWARF and show four (two technical, two scientific) example applications.





## 1  Introduction

Earth system models (ESMs), especially those including explicit atmospheric chemistry process descriptions, are very expensive in terms of required computing time and memory. As these models have been developed over a long time, they are not straight-forwardly efficiently applicable or even portable to new computing architectures. This problem is faced by most models of this kind around the world. Therefore, a strategy on how "to evolve weather and climate prediction models to next-generation computing technologies" (Müller et al., 2019) was developed within the HORIZON2020[1] funded ESCAPE

(Energy-efficient Scalable Algorithms for Weather Prediction at Exascale) project. The first point of this strategy is to "(i) identify domain-specific key algorithmic motifs in weather prediction and climate models (which we term Weather & Climate Dwarfs)" (Müller et al., 2019, introduction). "The dwarfs thus represent domain-specific mini-applications (Messer et al., 2018) which include direct input from the domain scientist together with documentation [...]" (Müller et al., 2019, Sect. 2.1). With the publication of Müller et al. (2019), the idea of restructuring comprehensive, established source codes, to separate them into

smaller pieces related to specifically simulated processes, in short the idea of formally modularising the source codes, became widely acknowledged in the community.

One work package of the Pilot Lab Exascale Earth System Modelling (PL-ExaESM) project of the German Helmholtz Association[2] was destined to tackle exactly this issue, i.e., to develop further the models in order to achieve efficient applicability on exa-scale architectures. The publication of Müller et al. (2019) inspired the work presented here.

Combining the basic design concepts of the Modular Earth Submodel System (MESSy, Jöckel et al., 2005), i.e. that each component is already coded as an individual submodel with the "dwarf strategy" developed and published by the ESCAPE project (Müller et al., 2019), resulted in the development of the MESSy DWARF model.

Its basic concept is to use MESSy datatypes and methods (for further details see Sects. 2 and 3) to provide the technical framework required to run one specific process implementation (a submodel in MESSy terminology) to be able to concentrate

in an efficient way on the optimisation of this specific process implementation or to "build" simplified models by combining a number of process implementations. To reach this goal, a model grid, the parallel decomposition, and a time control need to be provided from MESSy to replace settings usually provided by the dynamical models.

Before the DWARF development, MESSy already provided all datatypes and methods required to

– initialise the model variables by reading input data and transforming (remapping) it to the actual model grid,

– produce output, not only of instantaneous values, but also namelist driven standard diagnostic output as minima, maxima, averages etc.,

– control the time stepping and check pointing,

– perform a tendency analysis for prognostic variables.

---

[1]https://research-and-innovation.ec.europa.eu/funding/funding-opportunities/funding-programmes-and-open-calls/horizon-2020 (last access: 2024-06-04)

[2]The PL ExaESM lasted for October 2019 to September 2021, the successor project is named Joint Lab ExaEMS (JL-ExeESM), see https://www.exaesm.de/ (last access: 2024-06-04)



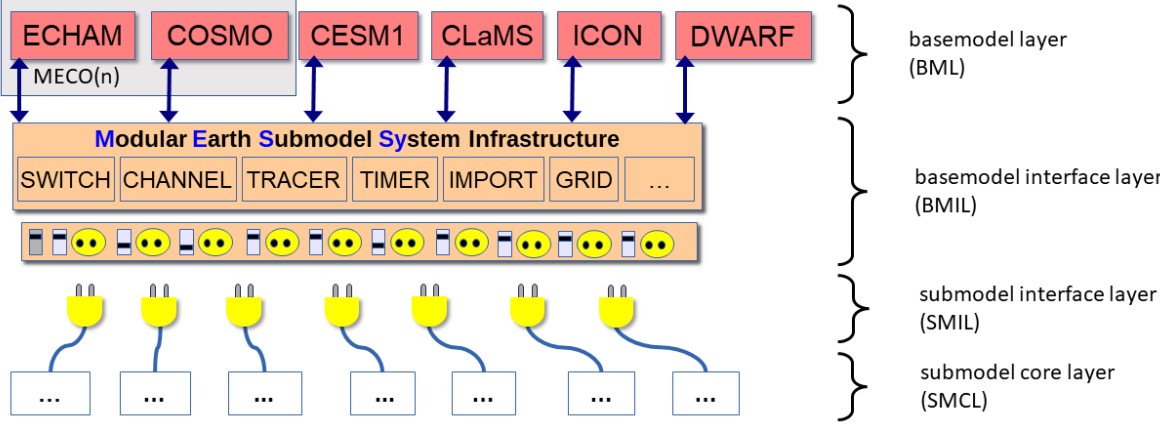

**Figure 1.** Sketch of the 4-layer code structure of MESSy. The image illustrates, how the different basemodels and the basemodel independent core layer of the individual submodels are interconnected via the two interface layers. The basemodel interface layer (BMIL) organises the correct workflow and data exchange between the regular submodels, which cores are connected by the submodel interface layer (SMIL) to the BMIL.

While the ESCAPE weather & climate dwarfs are seven specific dwarfs for seven process implementations, the MESSy DWARF adds a new basemodel to the MESSy framework, enabling to run each MESSy submodel (currently more than 90 are available and the number is growing) in this simplified setup. Each individual ESCAPE dwarf solves the problems by setting up its own environment (grid, parallelisation etc.), while the MESSy DWARF is a generic basemodel with which all process descriptions, available as MESSy submodels, can be applied. The ESCAPE dwarfs have the advantage, that the overhead is minimal, and thus providing the perfect setup to optimise this specific process. In contrast, the MESSy DWARF requires some additional overhead to the individually tackled submodel by using always the above listed MESSy funtionalities. However, while the higher overhead looks as a downside on the first glance, it is definitely an upside that the MESSy DWARF establishes a standard model, which can be used to drive all submodels, avoiding the need to develop individual drivers for each specific submodel. Furthermore, it is possible to run and test combinations of submodels with the DWARF. This is not only helpful for technical optimisations (see Sect. 4.4), but also opens up the DWARF for scientific use, e.g. for simplified models such as chemical box models (see Sect. 4.3).

Starting with a short introduction into the basic MESSy concept (Sect. 2), the following section provides an overview of the technical realisation of the MESSy DWARF (Sect. 3). Section 4 provides several examples of DWARF applications to illustrate its usability. Finally, the article is concluded with a short summary and a list of some future development plans (Sect. 5).



## 2 The MESSy infrastructure and submodel concept

The Modular Earth Submodel System (MESSy, Jöckel et al., 2005) is an integrated software framework, which was initially built to facilitate the implementation of atmospheric chemistry processes into existing models of atmospheric dynamics (Jöckel et al., 2010; Kerkweg and Jöckel, 2012). Its basic idea is that each process and diagnostics can be coded as an individual submodel. Such a submodel consists of two parts: the submodel core layer (SMCL) contains the implementation of the process coded in its basic entity (e.g., kinetics = box, convection = column process), while the submodel's interface layer (SMIL)

connects to the MESSy infrastructure components on the (basemodel dependent) basemodel interface layer (BMIL), which in turn is connected to the basemodel layer (BML), i.e. the basemodel (see Fig. 1). This 4-layer software architecture allows for the implementation of the SMCL to be completely independent of the basemodel, for instance its local variable dimensions, order of ranks, parallel decomposition, etc. . MESSy distinguishes between regular submodels and infrastructure submodels. While the regular submodels are realisations of individual processes (e.g., the kinetic solver for the gas phase chemistry, dry

deposition, convection asf.) or diagnostics, the MESSy infrastructure submodels establish the overall machinery required to orchestrate the individual submodels (data exchange, control flow, check-pointing, data import and export asf.). Thus, the infrastructure submodels establish the framework to execute the regular submodels. The MESSy infrastructure submodels comprise – among others – submodels for

–  memory management and output control (submodel CHANNEL, Appendix A[3], Jöckel et al., 2010),

–  import of data (submodel IMPORT, Kerkweg and Jöckel, 2015),

–  time and event control (submodel TIMER, Jöckel et al., 2010),

–  data exchange between basemodel and MESSy submodels (submodel DATA, Jöckel et al., 2005),

–  flow control, i.e. activation of submodels (submodel SWITCH), and calling of the submodels during simulation (CONTROL, Jöckel et al., 2005),

–  tracer data and tracer meta-data handling (submodel TRACER, Jöckel et al., 2008),

–  tendency diagnostic (submodel TENDENCY, Eichinger and Jöckel, 2014).

Table 1 lists the MESSy submodels mentioned in this publication including a very short description and respective references.

If MESSy is connected to a dynamical model, the dynamical model defines the grid, provides the parallel decomposition information including methods for data transpositions (e.g. the message passing interface (MPI) based subroutines), as well as

the set of prognostic variables and external data, such as land-sea-mask, surface elevation asf. . This information is picked up by the MESSy infrastructure submodels and translated into MESSy data types for harmonized usage by all MESSy submodels.

---

[3]Appendix A provides a very short introduction to the CHANNEL submodel. For readers not familiar with MESSy it introduces the most important terms used in the remainder of the article.





**Table 1.** List of MESSy submodels including a short description and citation (if possible).

| submodel name | description | citation |
|---|---|---|
| *infrastructure submodels* | | |
| CHANNEL | CHANNEL handles the memory and meta-data management, and data export. | Appendix A, Jöckel et al. (2010) |
| CONTROL | CONTROL controls the workflow of a MESSy simulations. In each entry point it calls the individual submodels (interlinked with SWITCH). | Jöckel et al. (2005) |
| DATA | DATA steers the data exchange between basemodel and MESSy submodels in both directions. | Jöckel et al. (2005) |
| DECOMP | DECOMP provides the parallel decomposition of the model grid. | this article |
| GRID | GRID provides the definition of the model grid and of grid related variables such as, longitude, latitude, box height, grid volume etc. | Kerkweg et al. (2018), this article |
| IMPORT | IMPORT performs the data import into the model. It contains three sub-submodels which themselves deal with different types of data. | Kerkweg and Jöckel (2015) |
| – IMPORT_GRID | IMPORT_GRID imports time series of gridded data in netCDF format to the actual model grid. | |
| – IMPORT_LT | IMPORT_LT performs the import of lookup tables. In MESSy they are used, e.g., to import optical properties required for the submodel AEROPT (Dietmüller et al., 2016), which calculates the AERosol OPTical properties. | |
| – IMPORT_TS | IMPORT_TS imports standard time series data. | |
| MPI | MPI establishes parallel communication (e.g., broadcasts, data gathering and scattering) via the Message Passing Interface (MPI). | |
| SWITCH | SWITCH activates individual submodels. | Jöckel et al. (2005) |
| TENDENCY | TENDENCY provides interfaces for the access and modification of prognostic variables. Furthermore, it enables the namelist driven possibility for tendency diagnostics for individual prognostic variables for single processes or collections of processes. | Eichinger and Jöckel (2014) |
| TRACER | TRACER manages tracers and their meta-data. This includes, among others, the definition of tracers, their attribution to specific processes (e.g., if a tracer should be advected) and the assignment of chemical and physical properties as molar mass, Henry coefficients etc. | Jöckel et al. (2008) |
| TIMER | TIMER establishes the time settings of the simulation and provides an event control, i.e., among others it sets the date components, schedules restarts, and provides the interfaces for scheduling of events in other submodels. | Jöckel et al. (2010) |
| *regular submodels* | | |
| DWARFDCD | DWARF specific submodel, defining a prognostic variable set for DWARF and enabling initialisation and nudging of the prognostic variables defined in this submodel | this article |
| JVAL | calculation of photolysis frequencies | Sander et al. (2014) |
| MECCA | kinetic solver for gas and liquid phase reactions | Sander et al. (2019) |
| ORBIT | calculation of orbital parameters | Dietmüller et al. (2016) |
| PTRACINIT | Passive TRACer INItialisation | Hofmann et al. (2016), this article: Appendix C |



## 3 The technical realisation of the MESSy DWARF

So far, MESSy was always connected to dynamical models, e.g. the global climate model ECHAM5 (Roeckner et al., 2006; Jöckel et al., 2010), or the regional weather and climate model COSMO (Rockel et al., 2008; Kerkweg and Jöckel, 2012). However, for simplified scientific applications or for technical tests (such as source code optimisation), a dynamical model is not always required or even counterproductive, due to unnecessary overhead, e.g. for performance analysis of a single MESSy submodel.

To run MESSy detached from legacy models, the MESSy infrastructure submodels need to provide everything what they usually inherit from the dynamical model. This comprises

1. the grid definition,

2. the parallel domain decomposition of the grid,

3. the methods for parallel communication and data transpositions (e.g. MPI based), and

4. the provision of external data (initial and boundary conditions).

Additionally, a regular submodel is required to provide what a dynamical core usually makes available, i.e., the definition, initialisation and modification of the prognostic variable set. This is necessary, to drive those MESSy submodels using prognostic variables as input. Thus, the new DWARF specific MESSy submodel DWARFDCD (DWARFs Dynamical Core Dummy) was developed to define a pseudo-prognostic variable set, even though no dynamical core is present (Sect. 3.1.1).

The following subsections provide more details on how the above listed missing features are implemented in the MESSy DWARF.

### 3.1 Data supply

In the usual MESSy model setup using a legacy model as basemodel, the MESSy submodels have access to two types of external data:

– data provided by the basemodel, and

– data imported by the MESSy submodel IMPORT.

The basemodel data itself separates into (i) initial and boundary data and (ii) basemodel variables.

(i) Initial data is used to initialise the model and therefore is only required at the very first model time step. In contrast to this, boundary data needs to be updated regularly, e.g. after discrete time intervals during the model simulation.

(ii) Basemodel variables are calculated by the basemodel. The most important variables are the prognostic variables, which are the variables of the solved equations of motion. The set of prognostic variables can therefore differ between basemodels. Non-prognostic variables are often called diagnostic variables.





```
!--------------------------------------------------------------------------------
! -*- f90 -*-
&CTRL
nudgedt_t  = 3600.0,
!nudgedt_u  = 3600.0,
/

&CPL
inp_tm1    = 'import_grid', 'inp3d_tm1',
! inp_qm1  = 'import_grid', 'inp3d_qm1',
inp_xlm1   = 'import_grid', 'inp3d_xlm1',
inp_xim1   = 'import_grid', 'inp3d_xim1',
! provide rhum instead of q
inp_rhum   = 'import_grid', 'inp3d_rhum',
inp_press  = 'import_grid', 'inp3d_press' ! required if inp_rhum is set
/
!--------------------------------------------------------------------------------
```

**Figure 2.** Example namelist file `dwarfdcd.nml` for the MESSy submodel DWARFDCD.

If MESSy is run in connection with a legacy basemodel, the basemodel specific variables are made available to the MESSy submodels by translation of those (in the BMIL) into so-called MESSy channel objects. The technical implementation of this translation in the BMIL depends on the basemodel. In EMAC, for instance, the translation is based on pointer association[4] of channel objects to the internal data structures of the ECHAM5 basemodel. In COSMO/MESSy, the COSMO source code has been modified to allocate memory of the COSMO variables directly in form of channel objects via the corresponding CHANNEL methods. Prognostic variables, specifically, are translated (i.e. made accessible) via the MESSy submodel TENDENCY (Eichinger and Jöckel, 2014). Diagnostic variables required by several MESSy submodels, but not present in the respective basemodel, are calculated in the (basemodel specific part of the) MESSy submodel DATA. One example might be the 3-dimensional geopotential height field, which is available in ECHAM5, but not in COSMO and therefore is declared and calculated in the COSMO specific part of DATA.

The DWARF, however, is a generic (basically "empty") basemodel without the functionality of a legacy basemodel. Thus, first a prognostic variable set needs to be defined. This is implemented within the submodel DWARFDCD (see Sect. 3.1.1). Second, all other variables in the DWARF become initial or boundary data and are imported by the MESSy submodel IMPORT.

Which set of prognostic variables is defined by the new MESSy submodel DWARFDCD and how they are modified, is described next (Sect. 3.1.1). The functionality of the IMPORT infrastructure submodel is explained in Sect. 3.1.2. The last subsection (Sect. 3.1.3) describes which variables are calculated additionally by the MESSy submodel DATA.

---

[4]as defined by the Fortran language standard





### 3.1.1 Definition and modification of prognostic variable

The MESSy infrastructure submodel TENDENCY (Eichinger and Jöckel, 2014) allows for detailed analyses of individual tendencies[5] of prognostic variables. Each process using a prognostic variable as input, or changing it, must access and/or modify

it via methods (i.e. calling of subroutines) of the infrastructure submodel TENDENCY. In this way, it can be ensured that the operator splitting concept is always followed (meaning implemented in a numerically correct way), and that all tendencies can be budgeted by TENDENCY, if desired.

If MESSy is connected to a dynamical model, the dynamical model determines the set of prognostic variables required to solve the equations of motion. However, the MESSy DWARF is (on purpose) not a dynamical model, but the MESSy

submodels are expecting specific prognostic variables to be accessible and modifiable via TENDENCY. Therefore, for the MESSy DWARF, the new submodel DWARFDCD defines the prognostic variables.

Currently, the following prognostic variables are taken into account for the MESSy DWARF:

- air temperature $t$ in K,

- the horizontal wind velocities $u$ and $v$ in m/s,

- the water vapour $q$ in kg/kg,

- the liquid water content $xl$ in kg/kg,

- the ice water content $xi$ in kg/kg.

Usually, the prognostic variables are updated by the dynamical model. Due to the configuration of the DWARF, the prognostic variables are only changed, if a MESSy submodel adds a tendency to the respective variable. However, it might be desirable

to initialise and change the content of the prognostic variables. Consequently, DWARFDCD provides an initialisation and a nudging option. Using the `&CTRL` namelist of the DWARFDCD namelist file (see Fig. 2) a relaxation time can be specified for each of the prognostic variables independently. The variable $nudgedt\_Y$, with $Y = t, u, v, q, xl, xi$, respectively, provides this relaxation time (in s), with which the respective variable is forced to follow the imported data, accessed via the corresponding `&CPL` namelist entries (`inp_Y`). The tendency ($tend\_Y$) added to the prognostic variable $Y$ is then calculated as

$$tend\_Y = (inp\_Y - start\_Y)/nudgedt\_Y \tag{1}$$

with $start\_Y$ being the current value of variable $Y$.

If only the initialisation, but no nudging, of a prognostic variable is required, the setting of the `inp_` parameters in the `&CPL` namelist is sufficient. A nudging coefficient must not be defined, as the definition of the nudging coefficients triggers the nudging.

---

[5]Tendency means here the time increment of a variable $X$: $X_{new} = X_{old} + \Delta X \times dt$ with $X_{new}$ indicating 2 time levels of the variable, $dt$ being the time step length and $\Delta X$ the tendency of the prognostic variable.





Note, a prognostic variable stays zero, if no input data is provided and no process adds a tendency to it. This is technically possible, as not all simplified DWARF setups require all prognostic variables.

As the DWARF will be used for measurement campaign analyses, a special case has been introduced for the initialisation and nudging of the water vapour mixing ratio. This renders useful, as measurements often provide only the relative humidity. Therefore the option was implemented to use the relative humidity (in %) as input for the forcing of the water vapour $q$ (in kg/kg) instead of prescribing the water vapour mixing ratio directly. This option is activated by providing `inp_rhum` instead of `inp_q`. Additionally, the conversion of relative humidity to specific humidity requires the pressure as input, thus `inp_press` needs to be provided in the `&CPL` namelist.

### 3.1.2  Initial and boundary data

One important part of each model simulation is the provision of initial and boundary data. The MESSy infrastructure submodel IMPORT provides the standard interface for data import at the beginning and during the model simulation. See the IMPORT User Manual (available in the supplement of Kerkweg and Jöckel (2015) or in the MESSy code distribution) for technical details and the usage of the IMPORT submodel including a detailed description of the meaning of namelist parameters. Two sub-submodels of IMPORT (compare Tab. 1 and  Kerkweg and Jöckel, 2015) are used for replacing basemodel functionalities:

- IMPORT_GRID

  The MESSy infrastructure submodel IMPORT_GRID imports 2- or 3-dimensional gridded data at the beginning of and during a simulation. Which data is imported and when is determined by namelists. The import process includes (i) reading in the requested data, (ii) distributing it according to the parallel decomposition, and (iii) remapping it onto the actual model grid used. The order of these steps depends on the basemodel used. For the DWARF and its limited-area grid (see Sect. 3.2), each parallel MPI task reads and interpolates the data for its part of the grid. The DWARF uses the SCRIP (A Spherical Coordinate Remapping and Interpolation Package) algorithm in IMPORT_GRID (Jones, 1999) for the remapping.

  Gridded data can be read once at the beginning of the simulation or in discrete time intervals during the simulation, if a time series of gridded data is available. The triggers for reading and the sequences of time steps to be selected from the files are defined by a Fortran namelist as user interface.

- IMPORT_TS

  IMPORT_TS provides an interface for reading standardised time series data, i.e., data available for a specific period in time and with a specific number of parameters, from ASCII or netCDF.

  In full 3-dimensional simulations this is e.g. used to read the vertical equatorial zonal wind profile for quasi biannual oszillation (QBO) nudging (Jöckel et al., 2016). In case of the DWARF, IMPORT_TS is especially helpful for box model simulations, i.e. for reading in measurement data to constrain the DWARF to the experiment conditions. Section 4.3 provides an example for this.





```
!------------------------------------------------------------------------------
! -*- f90 -*-

&CPL
inp_press  = 'import_grid', 'inp3d_press'
inp_pressi = 'import_grid', 'inp3d_pressi'
/
!------------------------------------------------------------------------------
```

**Figure 3.** Example namelist file `data.nml` for the MESSy submodel DATA.

### 3.1.3 DATA provision

The MESSy infrastructure submodel DATA serves as translator, providing the basemodel data to the MESSy submodels in a harmonized way. DATA performs different operations to simplify the access to the basemodel data for the submodels:

1. Variables available from different basemodels describing the same contents might be named differently. In this case, DATA defines a channel object reference (see App. A), to harmonize the channel and object name, with which the individual submodels access this variable.

    2. Variables, which are not directly available from the basemodel, but derivable from other variables, are calculated by DATA. For these, a channel object is defined (i.e. the respective memory is allocated), and the variable is calculated 210     during the time loop.

The first category is not available in the MESSy DWARF, as no basemodel is providing any data, i.e., the DWARF is completely driven by imported data. The second category is available for the DWARF.

Variables, which are usually provided by the dynamical model are required by some process submodels. If these can be calculated from the basic model setup and the prognostic variables, they are calculated within DATA for the DWARF. These 215 variables are:

    – the coriolis parameter $(1/\mathrm{s})$,

    – the vorticity $(1/\mathrm{s})$,

    – the density of dry air $(\mathrm{kg/m^3})$.

Even more important, the 3-dimensional pressure fields defined at the box centres or the vertical interfaces are required by 220 many calculations in the MESSy submodels. These can either be prescribed by imported data (in this case the imported data needs to be coupled to DATA via the `&CPL` namelist of DATA (see Fig. 3), or the pressure fields are allocated and calculated in DATA itself following the US standard atmosphere 1976 (U.S. Government Printing Office, 1976), using the height fields as determined in GRID (see Sect. 3.2).





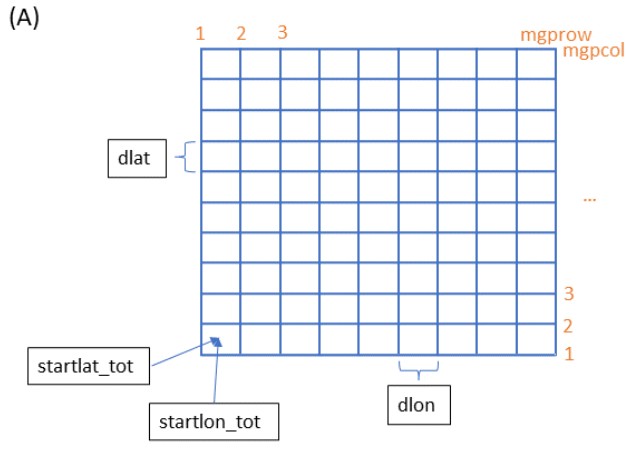
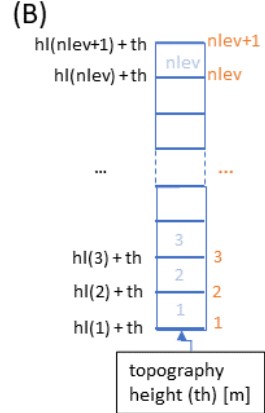

**Figure 4.** Illustration of the horizontal (A) and vertical (B) grid definition.

### 3.2 GRID_DEFinition

Usually, the grid is defined by the basemodel, and the MESSy sub-submodel GRID_DEF is used to provide / define all grid related variables required within the MESSy code. However, for the MESSy DWARF the grid itself needs to be defined in the MESSy infrastructure submodel GRID_DEF. For the start, we decided to use a very simple grid, as for technical tests only the number of grid boxes or the overall size of the grid matters. The basic grid layout is the same as used in the regional weather prediction and climate model COSMO (Doms and Baldauf, 2021; Rockel et al., 2008), however much more simplified, as the

grid defined for the DWARF in GRID_DEF does not allow for any rotation of the geographical grid.

The horizontal grid is defined as a rectangle spanning a limited-area (Fig. 4a), which is defined by the lower-left corner (the geographical coordinates of the grid box center point of the lower left grid corner have to be provided in the GRID_DEF namelist as `startlon_tot` in $[°E]$ and `startlat_tot` in $[°N]$), the number of grid boxes in longitude and latitude direction (`mgprow` and `mgpcol`), and the grid spacing (`dlon`/`dlat` in °E/°N, respectively). Figure 5 displays an example of

the namelist file of the GRID submodel `grid.nml`. The above listed namelist parameters are part of the `&CTRL_GRID_DEF` namelist.

The vertical grid (Fig. 4b) is dimensioned by the number of vertical boxes ($nlev$). Their vertical extent is defined via the namelist parameter $vc\_heighti$ (formula symbol $vc$), which provides the heights in m above topography ($H_{topo}$ in m) of the interface levels of the vertical grid boxes. The 3-dimensional fields containing the altitude above ground ($H_{gnd}^I$) and the altitude

above mean sea level ($H_{msl}^I$) at the vertical interfaces ($I$) of the grid are calculated from this input as follows:

$$H_{gnd}^I(i,j,k) = vc(k)$$
$$H_{msl}^I(i,j,k) = vc(k) + H_{topo}(i,j)$$

(2)





```
!--------------------------------------------------------------------------------
! -*- f90 -*-
&CTRL_GRID_DEF
!
mgpcol = 93
mgprow = 383
nlev   = 40
!
startlon_tot = 15.0
startlat_tot = 30.0
!
dlon = 0.5
dlat = 0.5
!
! height over ground for 40 levels:
vc_heighti = "22700.0, 20800.0, 19100.0, 17550.0, 16150.0, 14900.0, 13800.0, 12785.0,
              11875.0, 11020.0, 10205.0,  9440.0,  8710.0,  8015.0,  7355.0,  6725.0,
               6130.0,  5565.0,  5035.0,  4530.0,  4060.0,  3615.0,  3200.0,  2815.0,
               2455.0,  2125.0,  1820.0,  1545.0,  1295.0,  1070.0,   870.0,   695.0,
                542.0,   412.0,   303.0,   214.0,   143.0,    89.0,    49.0,    20.0,
                  0.0"
/
!--------------------------------------------------------------------------------
&CPL_GRID_DEF
inp_topoheight = 'import_grid', 'inp2d_topoheight',
inp_press_3d   = 'dwarf', 'press',
!lignore_mass = T,
/
!--------------------------------------------------------------------------------
```

**Figure 5.** Example namelist file grid.nml for the MESSy sub-submodel GRID_DEF.





```
!--------------------------------------------------------------------------
! -*- f90 -*-
&CTRL
npx = 4
npy = 2
/
!--------------------------------------------------------------------------
```

**Figure 6.** Example namelist file `decomp.nml` of the MESSy submodel DECOMP.

with $i, j$ and $k$ being the longitudinal, the latitudinal and vertical index, respectively. The height at the box mid-levels is calculated by

$$
\begin{aligned}
H_{gnd}(i,j,k) &= 0.5 * (H_{gnd}^I(i,j,k+1) + H_{gnd}^I(i,j,k)) \\
H_{msl}(i,j,k) &= 0.5 * (H_{msl}^I(i,j,k+1) + H_{msl}^I(i,j,k)).
\end{aligned}
\tag{3}
$$

The second namelist of the GRID namelist file (Fig. 5), i.e. `&CPL_GRID_DEF`, defines the coupling, i.e., the input required from other MESSy submodels. In the example case, the topography is read by the MESSy submodel IMPORT_GRID and the coupled object is named `inp_topoheight`. Additionally, the calculation of the air mass in the grid-box and the dry air mass in the grid-box requires the pressure. In Fig. 5 the pressure is not pre-described, but is calculated by the MESSy submodel DATA according to the US standard atmosphere (1976).

Last but not least, for a very reduced setup, when even the water vapour content is not known, the calculation of the grid mass can be skipped by setting the logical switch `lignore_mass` true.

### 3.3 DECOMPosition and parallel communication (MPI)

The MESSy infrastructure submodel DECOMP organises the parallel decomposition of the MESSy DWARF. The parallelisation is a classical horizontal domain decomposition along the longitudes and latitudes. The example namelist in Fig. 6 shows a
decomposition along the longitudinal range into 4 segments, and along the latitudinal range into 2 segments, requiring overall

**Table 2.** Process submodels used in example simulations.

| acronym | process submodels used |
| --- | --- |
| Orbit1/2 | ORBIT |
| Kin | ORBIT, JVAL, MECCA |
| Box1 | JVAL, MECCA |
| Box2 | JVAL, MECCA, PTRACINI |
| GPU | ORBIT, JVAL, MECCA |





**Table 3.** Basic setup of example simulations.

| acronym | lower left corner {°E, °N} | mgprow | mgpcol | nlev | dlon °E | dlat °N | start time | end time | time step seconds | output frequency minutes |
|---|---|---|---|---|---|---|---|---|---|---|
| | | | | | | | YYYY-MM-DD-HH | | | |
| Orbit1 | -150.0, -70.0 | 123 | 83 | 40 | 2.0 | 2.0 | 1998-03-19-23 | 1998-03-22-01 | 300 | 60 |
| Orbit2 | -150.0, -70.0 | 123 | 83 | 40 | 2.0 | 2.0 | 1997-12-31-23 | 1998-01-02-01 | 300 | 60 |
| Kin | -15.0, 30.0 | 63 | 43 | 40 | 0.5 | 0.5 | 1998-06-01-00 | 1998-06-02-01 | 120 | 10 |
| Box 1/2 | 6.4, 50.9 | 1 | 1 | 1 | 0.0001 | 0.0001 | 2021-01-10-10 | 2021-01-10-16 | 120 | 10 |
| GPU | -15.0, 30.0 | 63 | 43 | 40 | 0.5 | 0.5 | 1998-06-01-00 | 1998-06-01-01 | 120 | 10 |

4 x 2 = 8 MPI tasks. The number of grid boxes is divided by the number of tasks. If it cannot be divided without a remainder, the number of grid boxes is increased by one for as many tasks as correspond to the remainder. In our example, the latitude range covers 93 grid boxes, which are distributed among 2 tasks. In this case, the first task gets 46 grid boxes in latitudinal direction, while the second tasks gets 47 grid boxes. The same happens for the longitudinal range: $393 : 4 = 98.25$. Thus, the
first 3 tasks get 98 grid boxes, while the fourth task gets 99 grid boxes.

Internally, DECOMP saves the start indices of the respective local domains and determines the start longitude and latitude (`startlon`/`startlat`) of each local domain.

The parallel communication is established by the MESSy submodel MPI. It uses the message passing interface (MPI) to perform the communication between the different parallel tasks. The individual broadcast, gather, and scatter routines are
based on the respective implementation of the COSMO model.

## 4  Example applications

This article describes the design concept of the MESSy DWARF. The major driving force for developing the DWARF was on the one hand side the need to establish a standard tool to easily create simplified setups for technical tests. This is of major concern for the process of porting such a large code base as MESSy stepwise to GPU and for performance analysis of
individual submodels. On the other hand side, the possibility to easily develop simplified MESSy models in a unified way, e.g. box or column models, to focus on local processes without the need to consider horizontal transport, was still missing within the MESSy framework.

In the following, four examples for the applicability of the MESSy DWARF are presented. At the beginning a very simple box model calculating orbital variations (Sect. 4.1) is shown. A simplified 3-dimensional chemical model (Sect. 4.2) and a
chemical box model (Sect. 4.3) are presented next. The last example demonstrates, how a simplified 3-dimensional chemical model can be used for performance analysis in the framework of porting the chemical kinetics model to GPU (Sect. 4.4).



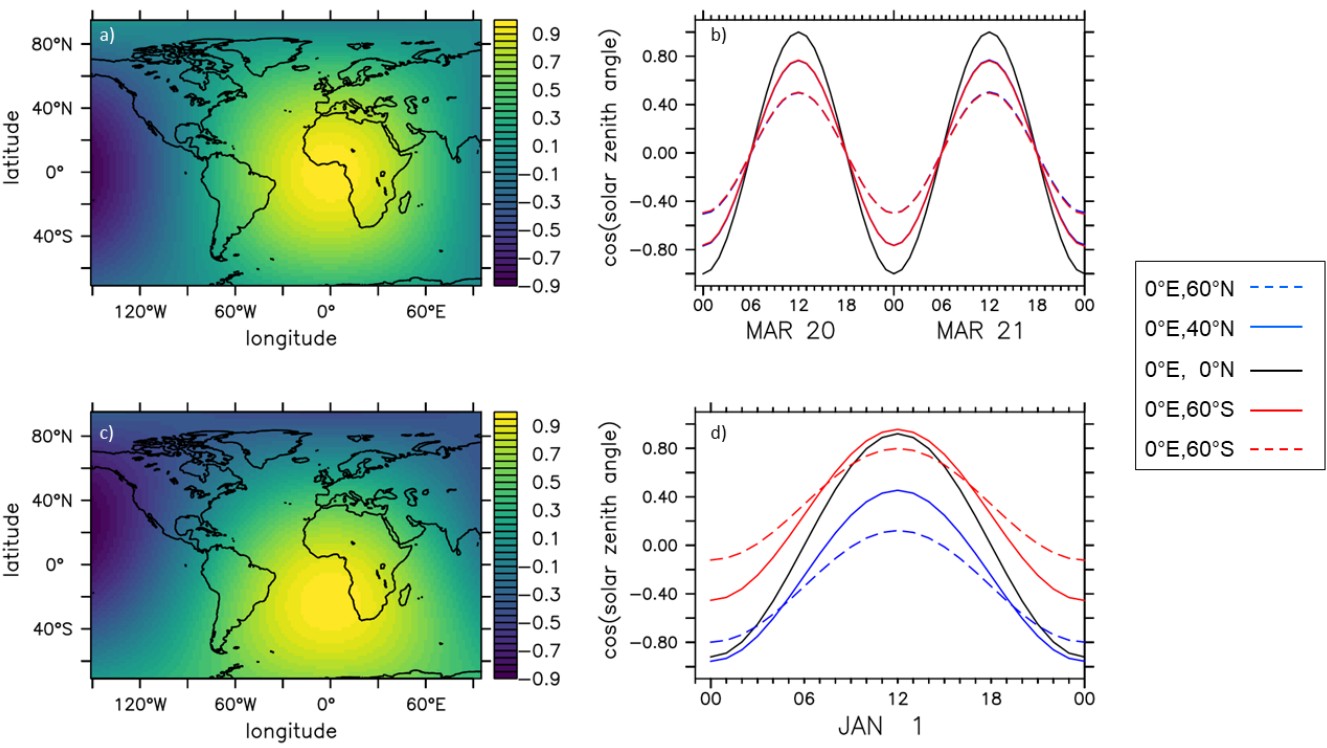

**Figure 7.** Cosine of the solar zenith angle calculated by the submodel ORBIT. Left: Area plots over the entire model domain for 20 March 1998, 12 UTC (upper row) and for 1 January 1998, 12 UTC (lower row). Right: Diurnal cycles of the cosine of the solar zenith angle 0° E and 60° S, 40° S, 0° N, 40° N and 60° N for the 20–21 March 1998 (upper) and 1 January 1998 (lower), respectively.

Table 2 lists the regular submodels used for the respective examples, while Tab. 3 provides an overview of the respective model configurations.

## 4.1 Orbital variations

As a very simple example, the submodel ORBIT (Dietmüller et al., 2016) calculating orbital parameters has been run for 20–21 March 1998 ("Orbit1" in Tabs. 2 and 3) and 1 January 1998 ("Orbit2"). Figure 7 shows some of the results.

    The diurnal cycle of the cosine of the solar zenith angle exhibits the largest amplitude at the Equator and the amplitude is decreasing with distance to the Equator (Fig. 7b,d). On equinox (20–21 March, Fig. 7b), the cosine of the solar zenith angle for 60° S and 40° S equals exactly those at 60° N and 40° N, respectively. However, this is only true for the equinox, while

differences in the cosine of the solar zenith angle are already visible before and after the equinox. In addition, in January (Fig. 7d) the shift of the solar zenith angle according to the season for the regions nearer to the pole is apparent.

    While the right panels give an impression of the modulation of the solar zenith angle from a local point of view, the left panels of Fig. 7 give a spatial impression. They present area-plots of the cosine of the solar zenith angle in the model domain

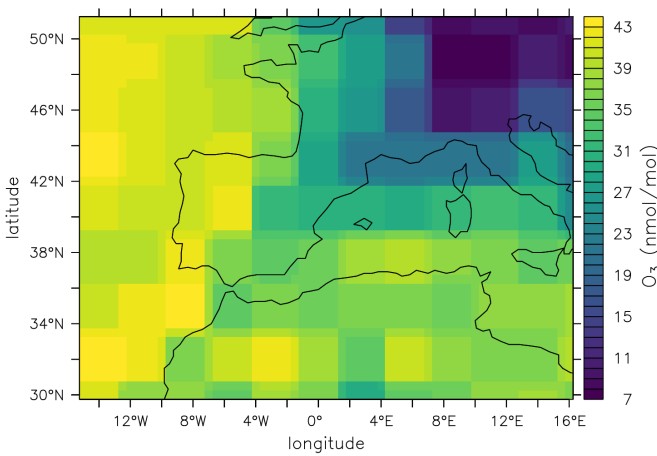

**Figure 8.** Ozone mixing ratio $(\mathrm{nmol\,mol^{-1}})$ in the lowest model layer 10 minutes after model start.

at 12 UTC. As expected, the maximum for March 20th is exactly at the Equator, while the maximum for January is shifted
towards the South Pole.

## 4.2    The chemical kinetics model

The second example ("Kin" in Tabs. 2 and 3) is a simple 3-dimensional setup for atmospheric gas phase chemical kinetics. The
only submodels used are (see Tab. 2):

–   the chemical kinetics submodel MECCA (Sander et al., 2005, 2019) solving the ordinary differential equation system
295        for gas phase chemistry, where the chemical mechanism is the same as the basic mechanism used by Jöckel et al. (2016),

–   the submodel JVAL (Sander et al., 2014) calculating the photolysis frequencies required within MECCA,

–   the submodel ORBIT (Dietmüller et al., 2016) providing the orbital parameters required by JVAL.

As no transport processes are included in this model setup, in fact, independent simulations in `mgprow x mgpcol x nlev`
`= 63x43x40 = 108360` grid boxes (compare Tab. 3) are performed. The boxes differ in their horizontal and vertical position
according to the defined grid. Differences in the simulated chemistry between the individual grid boxes are either due to the
initialisation of the chemical species or the prescribed external data.

For the initialisation of the tracers a restart file from a former T42 EMAC simulation is used. Figure 8 displays the initial
surface mixing ratio of ozone. The chessboard pattern results from the remapping of the coarse T42 (approx. 2.8° x 2.8°)
resolution to the 0.5° x 0.5° grid of the DWARF.
As the reaction rates in MECCA depend on temperature, pressure, and humidity, the initialisation of these profiles influences
the simulated kinetics directly. Note, that in this setup these profiles are only initialised. For the sake of simplicity they are
kept constant in time. Furthermore, the calculation of the photolysis frequencies (submodel JVAL) influences the chemistry



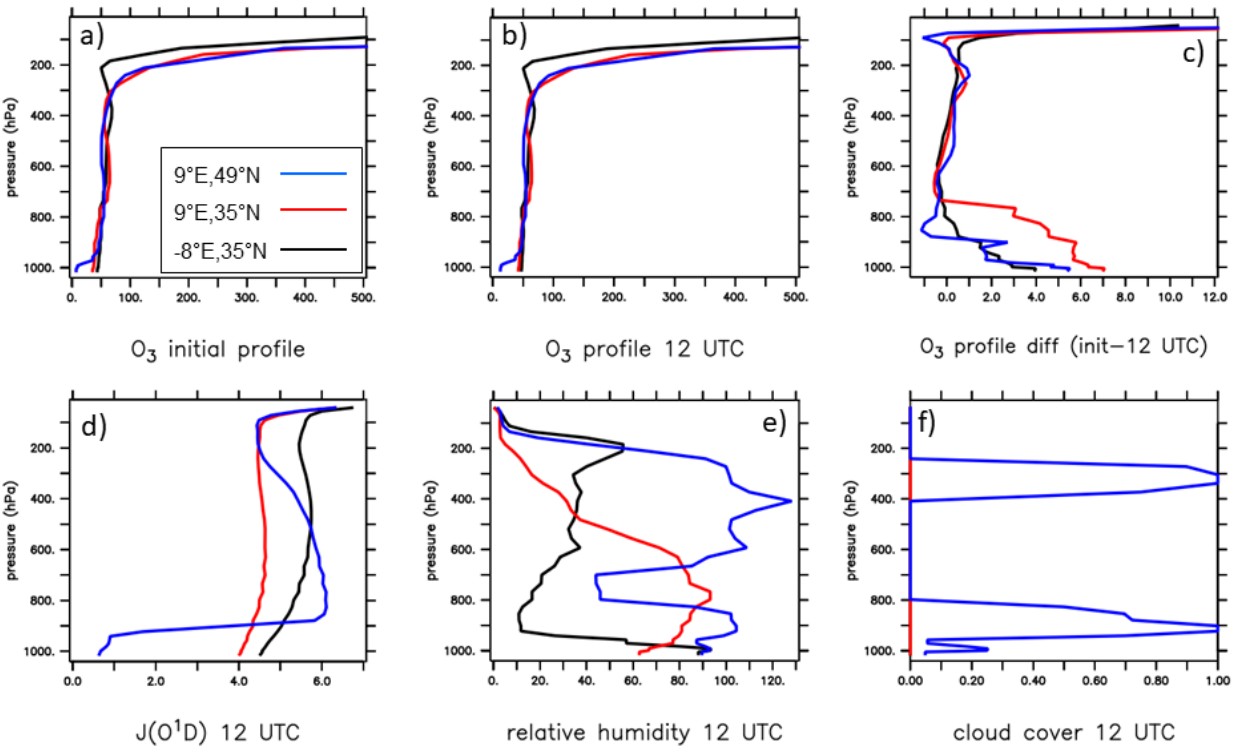

**Figure 9.** Profiles at three geo-locations are presented: blue: 9° E, 49° N; red: 9° E, 35° N and black: -8° E, 35° N.
The upper row displays ozone profiles ($\mathrm{nmol\,mol^{-1}}$) at model start (left), at 12 UTC (mid), and the difference between these profiles (right).
The lower row depicts profiles at 12 UTC of the photolysis frequency of $O^1D$ (in $10^5\,\mathrm{s^{-1}}$, left), of the relative humidity (in %, middle), and
of the cloud cover (as fraction between 0 and 1, right).

and depends itself on solar activity, orbital parameters, pressure, ozone, the cloud cover, the relative humidity, the albedo, and
on the surface type (land or sea).

To illustrate the spatial variation of these parameters, Fig. 9 displays vertical profiles at three different locations: blue: 9° E,
49° N; red: 9° E, 35° N and **black: -8° E, 35° N**. These locations have been chosen w.r.t. the surface ozone (compare Fig. 8),
for low, medium, and **high** surface ozone, respectively.

The upper row of Fig. 9 depicts the ozone profiles shortly after model start (left, annotated "initial"), at 12 UTC (middle),
and the difference of these profiles (right). Ozone is lowest in the lowest model layer, while it is some orders of magnitude

higher in the stratosphere (above approx. 250 hPa). However, the difference of the vertical profiles (right panel) between the
initial condition and the 12 UTC vertical profile exhibits different time evolutions of the profiles in each box. The bottom
row of Fig. 9 displays the respective vertical profiles of the photolysis frequency of $O^1D$ (left), the relative humidity (mid),
and the cloud cover (right). While the black and the red profiles are cloud free, the blue profile exhibits two distinct cloud


**Table 4.** Prescribed cosine of the solar zenith angle of the kinetic box model study (Box1), the hours at which the sun is "switched on" and "off", and the respective line colour in Fig. 10.

| cosine Zenith angle | start hour | end hour | line color |
|---|---|---|---|
| 0.9 | 11 | 14 | black |
| 0.7 | 11 | 14 | red |
| 0.5 | 11 | 14 | blue |
| 0.7 | 11 | 12 | light blue |

layers in which the 100 % relative humidity level is exceeded and supersaturation persists. Below the lower level cloud, the photolysis frequency is much smaller than for the two profiles without clouds. Above the cloud the photolysis frequency is slightly enhanced (compared to the other two profiles), caused by the reflection of sunlight on top of the underlying cloud. While the red and the blue profiles are above land, the black profile is above the ocean. However, when comparing the red with the black profile, the red profile exhibits the much higher relative humidity, except for the lowest model layer. In the pressure range approximately between 980 and 500 hPa, the red profile contains much more water vapour. This is reflected in

the photolysis frequencies, which are much lower for the red profile.

As atmospheric chemistry is a non-linear process, none of the shown aspects alone can explain the vertical profile of a specific chemical species, as here for example ozone. However, using a chemical box model provides an efficient tool to test dependencies on individual parameters. How such a box model can be build with the DWARF, and how it can be used, is explained in the next subsection.

### 4.3 The chemical box model

In contrast to the model setup described in the subsection above, this subsections deals with a single box model, i.e., `mgprow`, `mgpcol` and `nlev` are all 1, the bottom and the top height of this box are defined as `vc_heighti = 15.0,5.0`, i.e. the box mid is 10 m above the surface. The geographical location of the box is Jülich, Germany (50.9° N, 6.4° E). In the first example, only the submodels MECCA and JVAL are used. All required parameters are prescribed as time series data

via IMPORT_TS. The initialisation of the chemical species, if required, is taken from the same restart file as in the previous example. As photolysis frequencies depend on the light intensity, the MESSy submodel JVAL requires information about the ozone column above the simulated chemical box. This is no problem for a 3-dimensional dynamical model (additional information about the upper atmosphere ozone content is always input to JVAL anyway), however, if only one single box should be simulated, this profile information is missing. To enable the usage of JVAL also for single box model experiments,

JVAL has been expanded by the option to use an artificial profile as vertical column information. Appendix B provides further information about this expansion of JVAL.







**Figure 10.** Time evolution of different parameters and chemical species in a chemical box model simulation. Here, the cosine of the solar zenith angle (upper left panel) is the input parameter varied (see also Tab. 4).





**Table 5.** Emission times of OH and NO in the three simulations of the idealised kinetic box model study (Box2) and the respective line colour in Fig. 11.

| OH emission time (UTC) | NO emission time (UTC) | line color |
|---|---|---|
| 12 | 14 | black |
| 14 | 12 | red |
| 12 | 12 | blue |

The following two examples are shown to demonstrate how sensitivity studies could be easily performed with such a box model.

### 4.3.1 Sensitivity to solar radiation

Four simulations (Box1 in Tabs. 2 and 3) have been performed varying only one input parameter: the cosine of the solar zenith angle. Tables 3 and 4 provide an overview of the different simulations. The simulations were started at midnight in darkness. The sun was "switched on" at 11 UTC and switched off again at 14 UTC in three of the simulations, while in the simulation depicted in lightblue, darkness prevails already from 12 UTC on.

Exemplary for all, the photolysis frequency of the reaction $J(O^1D)$: $O_3 + h\nu \rightarrow O(^1D) + O_2$ is shown in Fig. 10. Natu-
rally, the photolysis frequency depends on the solar zenith angle, the higher the sun above the horizon, the larger the photolysis frequency. This artificially triggered onset and offset of photochemistry is mirrored in the time evolution of all nine chemical species depicted in the remaining nine panels. The higher the photolysis frequencies are, the faster ozone and methane decrease, while $H_2O_2$ and $HNO_3$ increase. The other species show a much more complicated time evolution, visualising the non-linear interactions in atmospheric chemistry.

### 355 4.3.2 Idealised experiment

The second example (Box2) presents a very idealised setup. In a simulation running from 10 UTC to 18 UTC, all chemical species except for ozone are initialised with zero. Ozone is initialised with a mixing ratio of $330 \, \text{nmol} \, \text{mol}^{-1}$. Photochemistry is "switched off", as the cosine of the solar zenith angle is set to $-1$. The simulation setups trigger OH and NO emissions at 12 UTC and / or 14 UTC. Table 5 provides an overview of the setups. The results (Fig. 11) show a clear dependence of the
production and destruction of the individual species on the emission times of NO and OH:

– Ozone is more efficiently destroyed by NO. If OH and NO are emitted at the same time, competing reactions lead to overall less ozone destruction (= larger ozone mixing ratio at the end of the simulation).



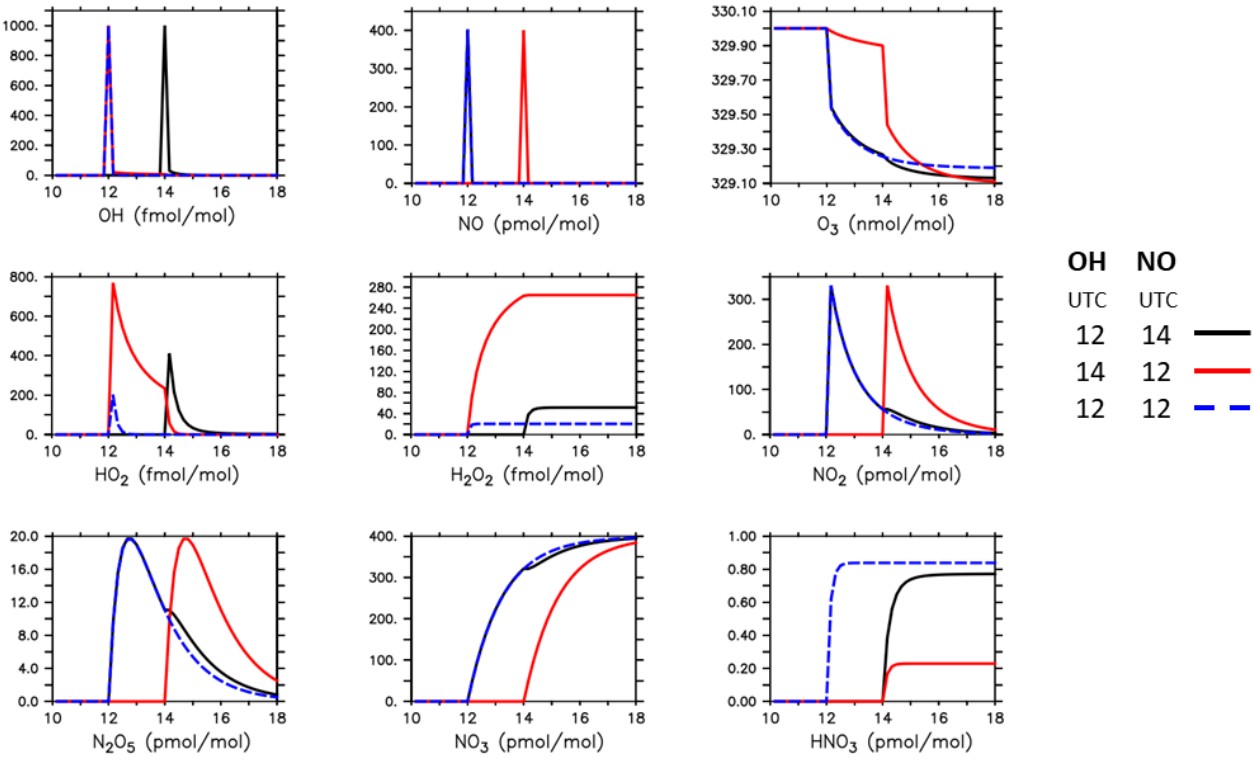

**Figure 11.** Time evolution of chemical species in a chemical box model simulation. Only ozone ($O_3$) was initialised. OH and NO were emitted at 12:00 and 14:00 in different order (see text and Tab. 5 for the respective line colours).

- If only OH is emitted first, $HO_2$ forms most efficiently, because nitrogen oxides do not compete with the $HO_2$ building reactions $O_3 + OH \rightarrow HO_2 + O_2$. Consequently, also $H_2O_2$ is build most efficiently, if OH is emitted first.

- Nitrogen oxides can only build, if NO is emitted. $NO_2$ is produced most efficiently, but also $NO_3$ and consequently $N_2O_5$ build up, if NO is emitted into the system.

- $HNO_3$ can only be build, if hydrogen and nitrogen are both added to the system. It forms most efficiently, if OH and NO are emitted simultaneously. By far less $HNO_3$ forms, if OH is added first, as in this case, most hydrogen ends up in $H_2O_2$, which cannot be converted to $HNO_3$ without photochemistry.

**4.4  Performance tests: GPU port of chemical kinetics**

This last example illustrates the usefulness of the DWARF for performance analysis. Analysing a specific submodel is much easier if the model configuration can be significantly simplified compared to complex basemodels. A smaller code base allows



**Table 6.** Runtime comparison between CPU and GPU for 2 chemical mechanisms (MIM and MOM) .

|  | overall runtime [s] | | speedup | kinetics runtime [s] | | speedup |
|---|---|---|---|---|---|---|
|  | CPU | GPU | | CPU | CPU | |
| MIM | 23.6 | 10 | 2.36 | 16.5 | 3.21 | 5.14 |
| MOM | 244.1 | 58 | 4.21 | 213.03 | 27.71 | 7.69 |

for a simpler and faster usage of performance analysis tools, as the time required to create the run time profiles is significantly reduced.

Here, the same simple 3-dimensional setup, as described in Sect. 4.2, was used to analyse and compare the run times of the MECCA submodel operated either entirely on CPUs or executing the chemical kinetic calculations on GPUs. The MEDINA tool (Alvanos and Christoudias, 2017; Christoudias et al., 2021) was used to create the CUDA code of the kinetics integration.

     Additionally, MECCA ran with 2 different reaction mechanisms: The standard mechanism (MIM, Mainz Isoprene Mechanism) taking into account 159 species and 360 reactions. The more complex mechanism based on MOM (Mainz Organic

Mechanism, Sander et al., 2019) has 729 species with 2193 reactions. Obviously, due to the larger matrix to be solved, it is expected, that the more complex mechanism shows even better performance (greater speedup) on GPU. Profiling was done with the NVIDIA Nsight Systems tool (https://developer.nvidia.com/nsight-systems last access 2024-03-25) on the JUWELS BOOSTER HPC system (Jülich Supercomputing Centre, 2021) by adding special markers to the DWARF code to enable the tracing of the code parts which are of interest here. The run times of the integration phase for these variants are displayed

in Tab. 6[6]. Additionally, Fig. 12 visualises the tracing displayed by the Nvidia Nsight Systems Tool for two subsequent time loop passes during the integration. Here, `mecca_physc` contains the call to the kinetic solver. Obviously, running the kinetic solver on GPUs speeds up the integration of the kinetics per se (Tab. 6, right), but also the overall run time (Tab. 6, left) is reduced by a substantial amount, i.e., a factor 2.36 and 4.21 for MIM and MOM, respectively. The more complex chemistry shows a much higher speedup, as the larger vector of chemical species results in more calculations, which can utilise better the

capability of the GPUs.

## 5    Summary and Outlook

The new MESSy generic basemodel DWARF is presented here. It was developed to provide the basis for a unified testing of individual MESSy submodels, for model performance analysis and improvement and, last but not least, as basis for setting up simplified models (e.g. box and column models) in a standardized way, by using all functionalities provided by the MESSy

framework.

     The examples presented in Sect. 4 of this article illustrate that the DWARF works, as intended, as simple process submodel test tool (Sect. 4.1), as 3-dimensional atmospheric chemistry multi-box model (Sect. 4.2), as 0-dimensional box model for

---

[6]The first timestep was excluded to avoid measuring the onetime GPU initialisation.

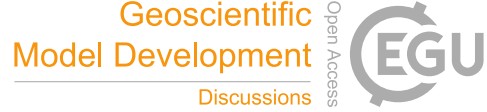

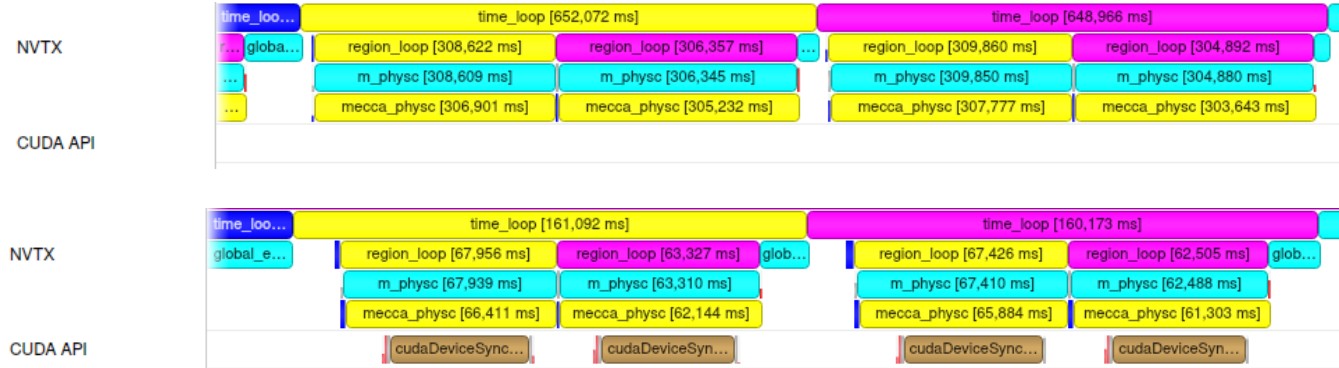

**Figure 12.** Example of an NVTX tracing with the Nvidia NSight System tool for the DWARF integration of the MIM chemical mechanism. Visualised are two subsequent time steps including the call of the kinetics integration: upper panel - CPU only, lower panel - kinetics integration on GPUs.

atmospheric chemistry sensitivity simulations (Sect. 4.3), and for performance tests when porting an individual submodel to GPUs (Sect. 4.4).

While these are basically technical further developments of the infrastructure of the DWARF, the DWARF has already been used for a larger technical development project in the framework of the natESM project (https://www.nat-esm.de/; last access: 2024-01-10). The aim of the project was to develop a concept for a MESSy infrastructure expansion to trigger copies between CPU and GPU memory most efficiently in MESSy setups, where only parts of the MESSy submodels are ported to GPU. To enable an easy start for the programmer from the natESM project, we developed a DWARF setup employing only

some artificial tracers and with five specialized submodels using the basic data handling and communication procedures of the MESSy infrastructure (i.e., from the TRACER, CHANNEL and TENDENCY infrastructure submodels). With this, the programmer could start relatively quickly with the actual code developments, while when using a comprehensive MESSy setup including a legacy dynamic model and atmospheric chemistry, it would have taken much longer for the programmer to understand the concepts and methods of the MESSy infrastructure submodels[7].

The MESSy DWARF is already in use for a scientific application. Within a PhD project (Do et al., 2024, in preparation), the chemistry box model DWARF setup as presented in Sect. 4.3 is further developed to be applicable for the analysis of experiments as e.g., performed within the atmospheric simulation chambers SAPHIR (Novelli et al., 2018; Franco et al., 2021), SAPHIR-STAR (Baker et al., 2023) and BATCH (Löher et al., 2024). This way, the improvements in the multiphase chemical kinetic model of MECCA, e.g. recent ones such as Soni et al. (2023); Wieser et al. (2023), can be applied and tested

readily in the MESSy framework for the global troposphere (Rosanka et al., 2024).

The implementation of the DWARF is at some points limited and has potential to be improved further. For example in future, the rather unflexible decomposition and corresponding MPI communication shall be performed via YAXT (Yet Another eXchange Tool, https://dkrz-sw.gitlab-pages.dkrz.de/yaxt/ (last access: 2024-01-10) ) to make it more flexible. Among other

---

[7]The development of the MESSy infrastructure expansion for GPUs will be published elsewhere, once it has been finalized.





advantages, this helps to get at least the MESSy DWARF code independent of license restricted software parts, which currently
prevent MESSy from becoming open source. In another ongoing project, the simple grid currently defined for DWARF will
be optionally replaceable by grids defined by the t8code library ( https://dlr-amr.github.io/t8code/ ; last access: 2024-01-10),
which features adaptive mesh refinement.

*Code and data availability.*  The Modular Earth Submodel System (MESSy, The MESSy Consortium (2023b)) is being continuously further
developed and applied by a consortium of institutions. The usage of MESSy and access to the source code is licenced to all affiliates of
institutions who are members of the MESSy Consortium. Institutions can become a member of the MESSy Consortium by signing the MESSy
Memorandum of Understanding. More information can be found on the MESSy Consortium website (http://www.messy-interface.org). The
reason for this license restriction is that many code parts of MESSy (and also its infrastructure) are adopted from license bound code,
which cannot be made available as open source easily. Especially for the DWARF, the DWARF specific parts of the MPI infrastructure
model have been adopted from the COSMO model, to which access is limited by a license. Anyhow, the MESSy consortium is working
on clarifying all licenses and providing MESSy under an open source licence. However, as this is a very tedious process, including many
institutions and different licenses, we can not wait with the publication of the concept of the DWARF until all license issues are clarified.
Especially, as the DWARF is already used for scientific and technical projects, and therefore should be citable. The code presented here was
developed based on MESSy version 2.55.2 and will be available in the next official release and is available and permanently archived at
zenodo (The MESSy Consortium, 2023a). Access to the code is granted to editors and reviewers.
Plot scripts and data displayed in the figures of this article are available in the supplement, as is a Manual for the MESSy DWARF.

## Appendix A:  The CHANNEL submodel

The MESSy infrastructure submodel CHANNEL provides data types and methods for the memory management and data
export within MESSy. It is described in detail by Jöckel et al. (2010) and the corresponding supplement. Here, we briefly
explain the terms used in the present manuscript. For further details, please refer to Jöckel et al. (2010).
CHANNEL organises the memory in so-called *channels*, which consist of the so-called *channel objects*. A *channel object* is
an individual data object comprising a pointer to the actual memory storing the contents of a variable and additional meta-data
such as a name and further attributes describing the object in more detail (e.g. units, descriptive long names, etc.). A *channel
object* moreover contains information about the geometry of the object, i.e., its spatial dimensions as well as information about
its parallel decomposition. To access a *channel object* two strings need to be specified, i.e., (i) the name of the channel and (ii)
the name of the *channel object* itself.

## Appendix B:  External vertical profile information in JVAL

As discussed in Sect. 4.3, JVAL requires the vertical profile of some variables. In a MESSy configuration coupled to a 3-
dimensional dynamical model, the column information is present and additional information about the ozone profile even
above the top of the dynamical model is imported via IMPORT_GRID (see Sander et al., 2014, for further details). If JVAL





is used in a single box model simulation, no such column information is available. To close this gap, JVAL was extended to use externally provided vertical profile information in box model applications. This mode is triggered by setting the `&CTRL` namelist parameter `l_artif_vprof` `.TRUE.`. In this case, the additional namelist `&CPL_PROF` is read:

```
&CPL_PROF
prof_nlev  = 19,
inp_prof_temp  ='import_ts','tempprof',
inp_prof_press ='import_ts','pressprof',
inp_prof_rhum  ='import_ts','rhumprof',
inp_prof_clp   ='import_ts','clpprof',
inp_prof_aclc  ='import_ts','aclcprof',
inp_prof_o3    ='import_ts','O3_H',
inp_prof_v3    ='import_ts','V3_H',
inp_prof_slf   ='import_grid','inp2d_slf',
/
```

`prof_nlev` defines the number of vertical levels. In the example, 19 levels are provided. The required information are
temperature (K), pressure (Pa), relative humidity (%), cloudiness (-), cloud cover (-), the ozone mixing ratio ($\mathrm{mol\,mol^{-1}}$) in the respective layer, and the ozone column above the respective level (in $\mathrm{molecules\,cm^{-2}}$). These data is read in via IMPORT_TS. Furthermore, the sea-land fraction is required as input. In the example this is provided via remapping of a 2-dimensional field by IMPORT_GRID.

## Appendix C: The MESSy submodel PTRACINI

The submodel PTRACINI (Prognostic TRACer INItialisation) was developed and used for stratosphere-troposphere transport diagnostics by Hofmann et al. (2016). In the meantime, the submodel was expanded. Most importantly, while Hofmann et al. (2016) prescribed the initial value with $10^{-7}$, the initial value can now be provided by namelist, either as constant or as channel object. The following overview of the submodel PTRACINI focuses on those functionalities required for the DWARF box model applications as presented in Sect. 4.3.

PTRACINI allows for the emission of tracers (defined elsewhere) dependent on prescribed criteria. The emission proceeds

- either once, at one single point in time, indicated by the parameter `TRINI(.)%ini1step`, or

- in regular intervals, if the interval is provided via an event, e.g., `TRINI(.)%EMIS_IOEVENT = 1,'hours','first',0.`

- The time of the (first) emission is defined via the parameter `TRINI(.)%EVENT_START`

See the TIMER manual available in Jöckel et al. (2010) for further information about events.
For the definitions of the criteria the following operators are available:

$$> , >= , < , <=$$

Figure C1 displays an example namelist file `ptracini.nml`, showing how the criteria can be defined:





```
!--------------------------------------------------------------------------------
! -*- f90 -*-
&CPL
inp_press = 'dwarf','press'

! SYNTAX for SET:
!INI1STEP:    - for onetime initialisation (if T, EMIS_IOEVENT is ignored,
!               if F, EVENT_START is ignored )
!EVENT_START: - set for onetime initialisation, is ignored for INI1STEP=F
!             - exact date to start the event (has to be a multiple of the
!               timestep, because exact time is prompted)
!EMIS_IOEVENT: - set for continous initialisation, is ignored for INI1STEP=T
!             - use step-intervall (otherwise oszillation for leapfrog scheme)
!IFNOT:       - if T, field is initialised when crits are NOT fullfilled
!
!TRACER: 'name_tr1', 'subname_tr1', 'name_tr2', 'subname_tr2', ...
!CRIT:   'channel', 'object', 'crit', const(_dp!) , 'compfld_cha','compfld_obj'

TRINI(1)%INI1STEP = T
TRINI(1)%EVENT_START = 2021,01,10,12,00,0
TRINI(1)%TRACER = 'NO','','','','','','','','','',
TRINI(1)%CRIT(1)='','','IF',4.e-10,'',''
!
TRINI(2)%INI1STEP = F
TRINI(2)%EVENT_START = 2021,01,10,14,00,0
TRINI(2)%EMIS_IOEVENT  = 1,'hours','first',0
TRINI(2)%TRACER = 'OH','','','','','','','','','',
TRINI(2)%CRIT(1)='','','IF',1.e-12,'',''
!
TRINI(3)%INI1STEP = F
TRINI(3)%EVENT_START = 2021,01,10,12,00,0
TRINI(3)%EMIS_IOEVENT  = 1,'steps','first',0
TRINI(3)%TRACER = 'HNO3','','','','','','','','','',
TRINI(3)%CRIT(1)='','','IF',1.e-12,'',''
!
TRINI(4)%INI1STEP = T
TRINI(4)%EVENT_START = 2021,01,10,12,06,0
TRINI(4)%IFNOT = F
TRINI(4)%TRACER = '4pvu','','','','','','','','','',
TRINI(4)%CRIT(1)='${MINSTANCE[$i]}','press','<=',90000,'',''
TRINI(4)%CRIT(2)='${MINSTANCE[$i]}','press','>=',15000,'',''
TRINI(4)%CRIT(3)='tropop','PV','>=',4,'',''
TRINI(4)%CRIT(4)='tracer_gp','qv','<=',0.001,'',''
/
!--------------------------------------------------------------------------------
```

**Figure C1.** Example namelist file `ptracini.nml` of the MESSy submodel PTRACINI.





1. In the first example ( `TRINI(1)` ) the tracer NO is initialised to the mixing ratio of $4\text{x}10^{-10}\,\text{mol}\,\text{mol}^{-1}$ at one single time step, i.e. on 10 January 2021, 12 UTC.

2. In the second example ( `TRINI(2)` ) the tracer OH is initialised and repeatedly reset to a mixing ratio of $10^{-10}\,\text{mol}\,\text{mol}^{-1}$. The initialisation takes place on 10 January 2021, 14 UTC and is reset to this value every hour.

3. The third example ( `TRINI(3)` ) is similar to the second example. The difference here is the definition of the event. By defining the event to `1,'steps'` the mixing ratio is reset every time step back to $10^{-10}\,\text{mol}\,\text{mol}^{-1}$.

4. The last example shows how criteria can be used. In this case the tracer $4\text{pvu}$ is initialised, if the pressure is between 900 and 150 hPa. Additionally, the potential vorticity (provided by the submodel TROPOP) needs to be larger than 4 PVU and the specific humidity has to be smaller than 0.001 $\text{kg}\,\text{kg}^{-1}$.

*Author contributions.* AK developed the overall DWARF concept, programmed it and wrote the largest part of the manuscript and the supplement. All authors contributed to the final stage of the manuscript. HDD further developed and tested the chemical box model. TK used the DWARF for GPU performance tests and wrote Sect. 4.4. SG tested the runability of the DWARF on the Juwels Cluster / Booster with different compilers. PJ and AK discussed the implementation of the DWARF according to the overall MESSy concept.

*Competing interests.* Astrid Kerkweg is an executive editor of GMD. Patrick Jöckel is topical editor of GMD. All other authors declare, that there are no competing interests

*Acknowledgements.* The MESSy DWARF was developed within the "Helmholtz-Inkubator Information & Data Science" project Pilot Lab Exascale Earth System Modelling (PL-ExaESM, https://www.fz-juelich.de/en/ias/jsc/projects/pl-exaesm : last access: 2024-03-25). This work used resources of the Deutsches Klimarechenzentrum (DKRZ) granted by its Scientific Steering Committee (WLA) under project ID 677.

The authors gratefully acknowledge the Earth System Modelling Project (ESM) for funding this work by providing computing time on the ESM partition of the supercomputer JUWELS (2021) at the Jülich Supercomputing Centre (JSC).



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
