# Peer review of "The MESSy DWARF (based on MESSy v2.55.2)"

_Geoscientific Model Development, 2024_

## Referee Comment (RC2)

Kerkweg et al. describe in their study the new basemodel DWARF, which is implemented in the MESSy infrastructure. DWARF is a simplified basemodel, which comprises the elementary contents of a basemodel. For example, DWARF defines a model grid, implements a time control (including the possibility of reruns), specifies the type of parallelization and data transfer (MPI) and give the possibility to create and initialize base variables. This makes it possible to perform simplified MESSy model simulations using DWARF as a basemodel instead of more comprehensive and time consuming GCMs such as ECHAM5, COSMO or ICON.

This has advantages for tasks, where the use of the comprehensive legacy base models lead to poor performance and at the same time not all the content of the used base model is required. For example the use of DWARF is for example, as the authors describe, useful in the case of porting of a small set of submodels to a new HPC architectures, for example to a GPU system, or in the case that you want only investigate local processes in a box or in a column.

The authors describe in their paper first the general infrastructure of MESSy and its workflow, also going into more detail on the MESSy submodels that are directly involved in this infrastructure. Then they present the technical realization and the design concept of DWARF and at the end the paper is completed with four application examples using DWARF.

In my opinion, the paper is very interesting and I can highly recommend it for publication. I think it's very good that the general infrastructure of MESSy is described first, and I see this as an added value of the paper, as it really helps to understand how the basemodel DWARF can be combined with MESSy and how it has to be set up. The description of the DWARF basemodel itself and how it works is comprehensively explained and understandable and therefore very useful if you want to use DWARF. I also think that the examples of what can you do with DWARF are sufficient and illustrate very nicely how DWARF can be used effectively.

Therefore I think that the paper is of great scientific importance and significance. Moreover it is written clearly and has a reasonable and understandable structure, language and figures. In my opinion the paper is already in an almost finished state. The paper complies with GMD guidelines and is fully suitable for publication in this journal.

**Remarks/Suggestions:**

A general comment from me concerns the abstract. I think it has partly more the form of an introduction. It refers very strongly to MESSy and in my eyes to less to DWARF itself. It should be underlined what advantages there are to use this new DWARF basemodel and to make the reader more curious to read the paper.

Specifically, I would shorten the description of MESSy in the abstract, make the description of DWARF more detailed, indicate which examples are discussed in the paper, and do a bit more advertising for the DWARF tool as a new useful and good application.

In total the paper is already in very good condition. I personally have found very few mistakes:

Line 8: "Since" → "For"

Line 80: "asf". Personally, I wouldn't use the abbreviation etc. as it's not necessarily familiar, but maybe I'm wrong.

Lines 95: "… set of prognostic variables …" → "… first set of prognostic variables …" ?

Line 120: Would you also consider nudging data as boundary data? If not, I would also mention it here.

Line 151: "… the prognostic variables." → "… the corresponding prognostic variables."

Line 211: ",as no basemodel is providing any data," → "as no data is provided to DWARF from another base model,"

Line 231: "Figure 5" → "Fig. 5"

Table 3: In Orbit1 and Orbit2 you have mgpcol=83 and dlat=2, with a start point of -70°N the last grid box would be at -70+(83*2)=96, that means at 96°N … Is that intentional?

Fig.7: "0°E, 60°S" →"0°E, 40°S" (red solid line)

Line 281: "Figure 7" → "Fig. 7"

Fig.8: "… 10 minutes after model start." → "10 minutes after model start (1.6.1998)."

Line 300: "… to the defined grid." → "… to the defined grid (from 30°N to 51.5°N and 15°W to 16.5°E)."

Line 302: "Figure 8" → "Fig. 8"

Line 305: "profiles" → "fields" ?

Line 306: "… these profiles …" → "… these temperature, pressure and humidity fields …". For better clarification (only a suggestion) …

Fig. 9:  1) "Profiles at three …" → "Ozone profiles at three …"
2) "… at model start (left) …" → "… at model start (0 UTC, 1.6.1998, left)"

Fig. 9 (panel top left): "$O_3$ initial profile" → "$O_3$ initial profile (0 UTC)"

Fig. 9 (panel top right): Is this really the difference between init – 12 UTC, or vice versa? I would rather expect the latter.

Line 317: "of $O^1D$" → "of $O^1D$ ($O_3$ + hv → $O(^1D)$ + $O_2$)

Line 352ff: I would suggest (but it´s only a suggestion): "The higher the photolysis frequencies are, the faster ozone decrease ($J(O^1D)$), OH increase ($O(^1D)$ + $H_2O$ → 2OH), $H_2O_2$ increase ($2HO_2$ → $H_2O_2$ + $O_2$), methane decrease ($CH_4$ + O(1D)), and $HNO_3$ increase ($NO_2$ + OH → $HNO_3$)."

Fig.11./Tab.5: There is something wrong. Corresponding to the panels in Fig.11 in the redline case OH is emitted at 12 UTC, and NO at 14 UTC, and in the blackline case OH at 14 UTC and NO at 12 UTC. But in Tab.5 and in the legend of Fig. 11 the corresponding times are reversed.

Line 367: "$HNO_3$ can only be build, …" → "$HNO_3$ can only be build ($NO_2$ + OH → $HNO_3$),…"

Line 381: "even better performance (greater speedup) on GPU" → "even better speedup on GPU". In my opinion MOM shows a better speedup, but not a better performance (because MIM is still faster).

Line 386: "Tab. $6^6$ ". I find this 6 as exponent from 6 confusing. Maybe you can change that somehow.

Line 401: "https://www.nat-esm.de/" → https://www.nat-esm.de

Line 410: "… is already in use …" →"… is also used …"

Line 422: "https://dlr-amr.github.io/t8code/ " → "https://dlr-amr.github.io/t8code "

---

## Author Comment (AC1)

**gmd-2024-117 – Reply to referee #1**

Dear referee #1,

Thank you very much for your supportive in-depth review, which has certainly helped to improved our manuscript considerably. Please find our replies to your comments below. Your original comments are repeated in italics, our replies – for easier reading – are depicted in blue, normal font, and text passages which we included in the manuscript are in bold.

*The authors of the article "The MESSy DWARF" describe The Modular Earth Submodel System (MESSy), a framework to integrate components to form an Earth system model. The topic of the article is an extension of the MESSy framework that simplifies the set-up and running of submodel(s) without the need for a full dynamical core to drive the simulation. Before this work, MESSy would always require a weather or climate forecast model as a "basemodel".*

*The underpinning concept of the presented approach is to substitute the basemodel and its role of providing the evolution of prognostic variables by a dummy component (the new submodel DWARFDCD). Prognostic variables are initialised to zero or, optionally, from input files and can be relaxed over time at a given rate. This approach provides other submodules with the relevant variables without the overhead of a basemodel, and its development is described as being inspired by the "dwarf strategy" developed and published in the ESCAPE project.*

*The authors highlight some differences to this dwarf strategy, namely the additional overhead incurred from always setting up a standard model with the entire coupling framework and infrastructure, which is notably more involved than the minimal drivers developed in ESCAPE, where each is bespoke to the corresponding component. This difference is being presented as an advantage because it allows to drive different components with the same setup. I do not fully subscribe to this point of view as the amount of required infrastructure is rather substantial, which can be a hindrance for technical exploration in experimental software stacks and the balance does not necessarily seem right for small submodels such as an individual microphysics parameterisation (one of the dwarves in ESCAPE). However, for larger submodels, e.g., related to atmospheric chemistry, which is a typical application scenario for MESSy, this seems less relevant, and overall I do agree that the gain in flexibility offers substantial advantages. In addition, one advantage that has not been listed explicitly, is the fact that the same submodel that has been developed and used with the DWARF can also be used without further changes coupled to a basemodel.*

In fact, we deliberately avoided the word 'advantage' and instead used the words 'downside' and 'upside', which in our understanding are less judgemental, to indicate that the size of the benefit or harm depends on the application. We adjusted the phrasing a bit. Thanks for the hint, that we did not mention the stability of the interfaces. We changed the text to

**While the larger overhead is definitely a downside for performance optimisation, it is an upside that the MESSy DWARF establishes a standard model, which can be used to drive all submodels, avoiding the need to develop individual drivers for each specific submodel. Furthermore, since the submodel interfaces are the same for all basemodels according to the MESSy concept, a submodel developed or optimised in a DWARF configuration can be used directly in a fully-fledged basemodel without any changes.**

*To me, the most substantial issue of the article is the structure and presentation of the concept and implementation. This starts already in the introduction, which does not sufficiently motivate the reasons why this development has been undertaken in the MESSy framework, and instead circles back-and-forth to the ESCAPE ideas and the similarities and differences. Pointing these out is valid but shouldn't constitute the basis of the work. Some of what I would have liked to read in this section comes in the first paragraph of Sec. 4 instead.*

Thanks for providing the hint, how to re-set the focus of the introduction. The revised introduction starts now with sentences similar to the first paragraph in Sect. 4, describes the basics of the DWARF concept and only after that relates to the idea of the ESCAPE project and the differences between the weather & climate dwarf concept and the MESSy DWARF concept. As we had to move major text blocks around, we do not cite the whole new

introduction here. Please have a look at the revised article.

*The introduction to the MESSy infrastructure and submodel concept in Sect. 2 is reasonably short and intuitive, with helpful pointers to additional resources for more details.*

As none of the referees expressed a need for changes in Sect. 2 we kept it, as it was.

*However, the description of the implementation in Sec. 3 dives immediately into technical details for individual infrastructure components of the MESSy framework. If this was instead motivated by the conceptual ideas of the dummy driver and the data requirements of a scientific submodel, it would benefit a wider audience beyond the users of the MESSy framework. This is notable, e.g., because more than once the relevant subsections start with statements similar to "usually this or that is defined/provided by the basemodel". In my opinion, if may suggest just one possibility: start with the description of the requirements of a hypothetical (or actual, as a case study) submodel component that is planned to be executed in a DWARF setting. Then explain how these inputs (e.g., prognostic variables), outputs (e.g., tendencies or diagnostic variables), and fundamental data structures (e.g., grid, domain decomposition), would be usually provided in the MESSy framework, thus highlighting the importance of the base-model for many of these. And ultimately, detail how the DWARF setup and DWARFDCD replace the basemodel for providing these, amended with technical details about their implementation in the basemodel interface layer.*

Thanks for this great idea. However, as the other two referees suggested that Section 3 is basically fine as it is, we decided to go with a compromise. The introduction of Sect. 3 now provides the information, what is required when using MECCA as the only regular submodel in a DWARF setup and how the input and output to / from MECCA is handled.

[revised manuscript text omitted]

- Sect. 4.3 and Sect. 4.4. do not require additional information, as this is contained in the previous examples or the information was already included.

*Finally, a few minor remarks to specific figures/parts of the manuscript:*

- *Figure 2 would benefit from some additional context in the caption, e.g., what nudging is active, what inputs are provided explicitly and which are omitted, etc. Moreover, I find the presence of commented namelist entries confusing (e.g., nudgedt_u).*

  Thanks for pointing that out. We cleaned up the figure and expanded the caption to:

  **Example namelist file `dwarfdcd.nml` for the MESSy submodel DWARFDCD. According to the &CPL namelist, all prognostic variables provided by DWARFDCD except $q$ are initialised from external data read by IMPORT_GRID. $q$ is initialised from an imported relative humidity field. For this calculation, additionally, the pressure field is required, which is also provided by IMPORT_GRID. The only time-varying, i.e., nudged prognostic variable is the temperature, as only for this a nudging coefficient is defined in the &CTRL namelist.**

- *The labeling inside Figure 4 suggests that mgprow/mgpcol denote the number of grid points per direction, while the text in Sec. 3.2 refers to them as the number of grid boxes (which would imply the number of grid points is $mgprow + 1/mgpcol + 1$).*

  We corrected the figure.

- *The description of the decomposition in Sec. 3.3 is misleading: For example, "the latitude range covers 93 grid boxes, which are distributed among 2 tasks" is factually wrong, because these are distributed among 8 tasks. The latitude range is rather decomposed into two subranges, each of which is assigned to one of the two sets of compute tasks in latitudinal direction. The corresponding longitudinal range in each of these sets is then further subdivided and assigned to one of the four compute tasks assigned to the latitudinal subrange.*

  Thanks for spotting this misleading formulation. We used the word "segments" before, therefore we reformulated this paragraph:

  **In our example, the latitude range covers 93 grid boxes, which are distributed among 2 segments. In this case, the tasks of the first segment get 46 grid boxes in latitudinal direction, while the tasks in the second segment get 47 grid boxes. The same happens for the longitudinal range: $393 : 4 = 98.25$. Thus, the tasks in the first three segments get 98 grid boxes, while the tasks in the fourth segment get 99 grid boxes. Thus the task 0 gets $98x46$ grid boxes, while task 7 gets $99x47$ grid boxes.**

- *The technical description of the performance benefits of the GPU port is incomplete. Performance numbers are presented without stating the used hardware (the mentioning of JUWELS-BOOSTER for the GPU numbers suggests that A100 GPUs were used but it is entirely unclear how many and to what CPUs this is being compared).*

  We added the missing information:

  **On this machine one compute node equipped with 2 AMD EPYC 7402 processors and 4 NVIDIA A100 40GB was used. For the CPU run the GPUs were disabled.**

- *The language is in some places rather informal, for example the use of "Anyhow" in the Code and data availability statement.*

  We removed the "anyhow" in the code availability section. However, detecting these "informal" language in additional parts is not so easy for non-native speakers. We hope that the "anyhow done" copy-editing will help in that respect.

*Overall, the presented work is substantial, the concepts appear sound and the benefits and applicability promising. But the presentation in the article would benefit from a clearer structure and storyline.*

Thanks again for this overall very positive review.

---

## Author Comment (AC2)

**gmd-2024-117 – Reply to referee #2**

Dear referee #2,

Thank you for your supportive review. Please find our replies to your comments below. Your original comments are repeated in italics, our replies – for easier reading –in blue, normal font, and text passages which we included in the manuscript are in bold.

*Kerkweg et al. describe in their study the new basemodel DWARF, which is implemented in the MESSy infrastructure. DWARF is a simplified basemodel, which comprises the elementary contents of a basemodel. For example, DWARF defines a model grid, implements a time control (including the possibility of reruns), specifies the type of parallelization and data transfer (MPI) and give the possibility to create and initialize base variables. This makes it possible to perform simplified MESSy model simulations using DWARF as a basemodel instead of more comprehensive and time consuming GCMs such as ECHAM5, COSMO or ICON.*

*This has advantages for tasks, where the use of the comprehensive legacy base models lead to poor performance and at the same time not all the content of the used base model is required. For example the use of DWARF is for example, as the authors describe, useful in the case of porting of a small set of submodels to a new HPC architectures, for example to a GPU system, or in the case that you want only investigate local processes in a box or in a column.*

*The authors describe in their paper first the general infrastructure of MESSy and its workflow, also going into more detail on the MESSy submodels that are directly involved in this infrastructure. Then they present the technical realization and the design concept of DWARF and at the end the paper is completed with four application examples using DWARF.*

*In my opinion, the paper is very interesting and I can highly recommend it for publication. I think it's very good that the general infrastructure of MESSy is described first, and I see this as an added value of the paper, as it really helps to understand how the basemodel DWARF can be combined with MESSy and how it has to be set up. The description of the DWARF basemodel itself and how it works is comprehensively explained and understandable and therefore very useful if you want to use DWARF. I also think that the examples of what can you do with DWARF are sufficient and illustrate very nicely how DWARF can be used effectively.*

*Therefore I think that the paper is of great scientific importance and significance. Moreover it is written clearly and has a reasonable and understandable structure, language and figures. In my opinion the paper is already in an almost finished state. The paper complies with GMD guidelines and is fully suitable for publication in this journal.*

Thank you very much for this positive assessment of our work!

*Remarks/Suggestions: A general comment from me concerns the abstract. I think it has partly more the form of an introduction. It refers very strongly to MESSy and in my eyes to less to DWARF itself. It should be underlined what advantages there are to use this new DWARF basemodel and to make the reader more curious to read the paper.*

*Specifically, I would shorten the description of MESSy in the abstract, make the description of DWARF more detailed, indicate which examples are discussed in the paper, and do a bit more advertising for the DWARF tool as a new useful and good application.*

As suggested, we shortened the original abstract and added more details about DWARF to it:
**Adaptation of Earth system model (ESM) codes to modern computing architectures is challenging, as ESMs consist of a multitude of different components. Historically grown and developed by scientists rather than software engineers, the codes of the individual components are often interwoven, making the optimisation of the ESMs rather challenging, if not impossible. Thus, in the last years the codes became increasingly modularised and with that, different components are disentangled from each other. This helps porting the code section by section to modern computing architectures, e.g. to GPUs.**

**Since more than 20 years, the modularisation is the fundamental concept of the Modular Earth Submodel System (MESSy). It is an integrated framework providing data structures and methods to build comprehensive ESMs from individual components. Each component, e.g., a cloud**

microphysical scheme, dry deposition of trace gases, or diagnostic tools, as output along satellite orbits, is coded as an individual, so-called submodel. Each submodel is connected via the MESSy infrastructure with all other components, together forming a comprehensive model system. MESSy was mainly developed for research in atmospheric chemistry, and so far it is always connected to a dynamical (climate or weather forecast) model, what we call basemodel. The basemodel is a development outside the MESSy framework. Running a full dynamical model for technical tests when porting only one submodel is a tedious task and unnecessarily resource consuming. Therefore, we developed the so-called MESSy DWARF, a simplified basemodel based on the MESSy infrastructure. We implemented the definition of a very simple grid, parallelisation scheme, and a time control to replace a fully-fledged basemodel.

The MESSy DWARF serves as a valuable tool for technical applications, such as porting individual component implementations to GPUs and performance tests, or as easy test environment for process implementations. Due to the MESSy structure, the applied components can be used in a dynamical model without any changes, because the interface is exactly the same. Furthermore, the MESSy DWARF is suited for scientific purposes running simplified models (with only a selection of components), e.g., a chemical box model for the analysis of chamber experiments, or a trajectory box model imitating an air parcel rising slowly into the stratosphere. Column and plume models could also easily be build based on the DWARF.

In this article we introduce the technical setup of the MESSy DWARF and show four example applications: (1) a simple application using a component calculating orbital parameters, (2) a chemical kinetics model including photolysis frequencies calculation, (3) an application of a chemical box model, and (4) some details on a GPU performance test of the chemical kinetics model.

*In total the paper is already in very good condition. I personally have found very few mistakes:*
*Line 8: "Since" → "For"*

Done

*Line 80: "asf". Personally, I wouldn't use the abbreviation etc. as it's not necessarily familiar, but maybe I'm wrong.*

Changed to etc.

*Line 120: Would you also consider nudging data as boundary data? If not, I would also mention it here.*

Yes, in our understanding nudging data is also some kind of boundary data. To clarify this, we added it in this sentence:
**The basemodel data itself separates into (i) initial and boundary data (including nudging data) and (ii) basemodel variables.**

*Line 211: ",as no basemodel is providing any data," → "as no data is provided to DWARF from another base model,"*

We do not agree with the suggested re-wording, as DWARF also means the basemodel itself. We chose the following rephrasing to make it clearer:
**The first category is not available in the MESSy DWARF, as in the DWARF set-up there is no data-providing basemodel, i.e., the DWARF is completely driven by imported data. The second category is available from the DWARF.**

*Line 231: "Figure 5" → "Fig. 5"*

The GMD guideline (`https://www.geoscientific-model-development.net/submission.html`) states the following: "The abbreviation "Fig." should be used when it appears in running text and should be followed by a number unless it comes at the beginning of a sentence, e.g.: "The results are depicted in Fig. 5. Figure 9 reveals that..."." . Therefore we keep Figure 5 ...

*Table 3: In Orbit1 and Orbit2 you have mgpcol=83 and dlat=2, with a start point of -70°N the last grid box would be at -70+(83\*2)=96, that means at 96°N ... Is that intentional?*

No. Thanks for spotting this. We re-run the simulation with 71 grid boxes for the latitude and changed the paper accordingly.

*Fig.7: "0°E, 60°S" →"0°E, 40°S" (red solid line)*

Thanks for spotting this. We corrected it.

*Line 281: "Figure 7" → "Fig. 7"*

Same as for Figure 5 above.

*Fig.8: ". . . 10 minutes after model start." → "10 minutes after model start (1.6.1998)."*

Date added.

*Line 300: ". . . to the defined grid." → ". . . to the defined grid (from 30°N to 51.5°N and 15°W to 16.5°E)."*

We added the requested information.

*Line 302: "Figure 8" → "Fig. 8"*

Kept. Reason see above

*Line 305: "profiles" → "fields" ?*

Changed.

*Line 306: ". . . these profiles . . . " → ". . . these temperature, pressure and humidity fields . . . ". For better clarification (only a suggestion) . . .*

Done. This will also clarify one remark by referee # 3.

*Fig. 9: 1) "Profiles at three . . . " → "Ozone profiles at three . . . "*

Done.

*2) ". . . at model start (left) . . . " → ". . . at model start (0 UTC, 1.6.1998, left)"*

Changed.

*Fig. 9 (panel top left): "O3 initial profile" → "O3 initial profile (0 UTC)"*

Changed.

*Fig. 9 (panel top right): Is this really the difference between init – 12 UTC, or vice versa? I would rather expect the latter.*

You are right. Thanks for spotting this. We changed the annotation accordingly.

*Line 317: "of O1D" → "of O1D (O3 + hv → O(1D) + O2)*

Added.

*Line 352ff: I would suggest (but it´s only a suggestion): "The higher the photolysis frequencies are, the faster ozone decrease (J(O1D)), OH increase (O(1D) + H2O → 2OH), H2O2 increase (2HO2 → H2O2 + O2), methane decrease (CH4 + O(1D)), and HNO3 increase (NO2 + OH → HNO3)."*

Thanks for the suggestion. We changed it to

**The higher the photolysis frequencies are, the faster**

- **ozone decreases (mainly due to** $O_3$ + h$\nu$ → $O(^1D)$ + $O_2$ **),**

- **OH increases (mainly driven by** $O(^1D)$ + $H_2O$ → $2OH$ **),**

- **$H_2O_2$ increases (mainly due to** $2HO_2$ → $H_2O_2$ + $O_2$ **),**

- **methane decreases (mainly driven by** $CH_4$ + OH → $CH_3$ + $H_2O$ **), and**

- **$HNO_3$ increases (mainly due to** $NO_2$ + OH → $HNO_3$ **).**

*Fig.11./Tab.5: There is something wrong. Corresponding to the panels in Fig.11 in the redline case OH is emitted at 12 UTC, and NO at 14 UTC, and in the blackline case OH at 14 UTC and NO at 12 UTC. But in Tab.5 and in the legend of Fig. 11 the corresponding times are reversed.*

Thanks for discovering this mistake. We corrected the table and the legend accordingly.

*Line 367: "HNO3 can only be build, . . . " → "HNO3 can only be build (NO2 + OH → HNO3),. . . "*

Reaction added as suggested.

*Line 381: "even better performance (greater speedup) on GPU" → "even better speedup on GPU". In my opinion MOM shows a better speedup, but not a better performance (because MIM is still faster).*

"even better performance (greater speedup) on GPU" Changed to "greater speedup on GPU"

*Line 386: "Tab. 66 ". I find this 6 as exponent from 6 confusing. Maybe you can change that somehow.*

We moved the footnote to "run times" to avoid this confusion.

*Line 401:* "*https://www.nat-esm.de/*" → *https://www.nat-esm.de*

Slash was removed.

*Line 410:* "*... is already in use ...*" →"*... is also used ...*"

Changed.

*Line 422:* "*https://dlr-amr.github.io/t8code/* " → "*https://dlr-amr.github.io/t8code* "

Slash was removed.

---

## Author Comment (AC3)

**gmd-2024-117 – Reply to referee #3**

Dear referee #3,

Thank you for your supportive review. Please find our replies to your comments below. Your original comments are repeated in italics, our replies in blue, normal font, and text passages which we included in the manuscript are in bold.

*The paper describes the implementation of the submodel DWARF within the framework of the modular earth sub-model system (MESSy) and its application in some exemplified test cases. The DWARF submodel allows the creation of a simplified model substituting the usually used dynamical base model in the MESSy framework. The advantage of this concept is having a simple test environment for the application, development, and performance testing of single submodels or a combination of submodels.*

We like to clarify here, that the MESSy DWARF is not a submodel but a basemodel of the MESSy system. A submodel alone could never replace a basemodel.

*Overall it is a well-written paper, well suited for GMD and should be published after minor revisions.*

Thanks for your positive assessment of our work.

*General comment:*

*Describe in more detail the difference between DWARF and DWARFDCD at the beginning of Section 3. Maybe describe it in more detail, e.g., as done in the supplement on page 10.*

Comparing DWARF to DWARFDCD means to compare a basemodel with a regular submodel, respectively. DWARFDCD is a MESSy submodel required in DWARF setups to provide the prognostic variables, which would be provided by the basemodel, if a fully-fledged dynamical model is used. Following the suggestion of referee #1 we added, at the beginning of Sect. 3, what would be required, if the submodel MECCA should be run in the DWARF setup. This includes the following paragraph:
**Temperature and specific humidity are prognostic variables. Dynamical basemodels provide the set of prognostic variables as determined by their dynamical core. In MESSy, these are made accessible to the MESSy submodels via the infrastructure submodel TENDENCY (see Sect. 3.1.1). As the DWARF does not contain a dynamical core, the regular MESSy submodel DWARFDCD (DWARF's Dynamical Core Dummy) provides a set of prognostic variables (see Sect. 3.1.1).**
This hopefully clarifies the relationship between DWARF and DWARFDCD.

*Section 3:*

*Line 124 and line 149:*

*You mention the prognostic variables provided by the base model and restrict this to these of the equation of motion. It would be best if you wrote this more generally, as all these variables are integrated forward from the primitive equations.*

changed in line 124 to**.. which are the variables used to solve the primitive equations.**
line 149 was deleted due to changes made in response to referee # 1.

*Section 4:*

*Line 306/307: "For the sake of simplicity they are kept constant in time."*

*Does that mean that the tendencies are not added to the chemical tracers?*

*Please clarify.*

No it does not. This paragraph is not at all referring to tracer tendencies. It is about temperature, pressure and specific humidity fields, which are required for the calculation of the reaction rates. Due to a change w.r.t. a comment by referee #2, this should be clearer now. The whole paragraph now reads:
**As the reaction rates in MECCA depend on temperature, pressure, and humidity, the initialisation of these fields influences the simulated kinetics directly. Note, that in this setup the temperature, pressure and humidity fields are only initialised. For the sake of simplicity they are kept constant over time. Furthermore, the calculation of the photolysis frequencies (submodel JVAL) influences**

**the chemistry and depends itself on solar activity, orbital parameters, pressure, ozone, the cloud cover, the relative humidity, the albedo, and on the surface type (land or sea).**

*Technical correction:*

*Avoid setting an extra period in cases where the sentence ends with an abbreviation: Line 78, Line 95*

Changed.

*Change for clarity: Line 137:*

*. . . a prognostic variable set . . .  → . . . a set of prognostic variables . . .*

Done.

---

## Author Response (AR2)

Dear Editor,

thanks for accepting the manuscript "as is"!

Best regards, Astrid Kerkweg

Dear production office team,

No further replies are required, as the manuscript was accepted "as is".

We removed the colour from the text of the manuscript, as requested by the production office. Anyhow, it is a pity, that this does not work, because we think this makes the description of the manuscript easier to follow (otherwise one reads black test telling the line is blue or similar).

Best regards, Astrid Kerkweg